# Site-Specific Evolutionary Rate Shifts in HIV-1 and SIV

**DOI:** 10.3390/v12111312

**Published:** 2020-11-16

**Authors:** Maoz Gelbart, Adi Stern

**Affiliations:** The Shmunis School of Biomedicine and Cancer Research, George S. Wise Faculty of Life Sciences, Tel Aviv University, Tel Aviv 6997801, Israel; maozgelbart@mail.tau.ac.il

**Keywords:** phylogenetics, rate shifts, cross-species transmission, HIV

## Abstract

Site-specific evolutionary rate shifts are defined as protein sites, where the rate of substitution has changed dramatically across the phylogeny. With respect to a given clade, sites may either undergo a rate acceleration or a rate deceleration, reflecting a site that was conserved and became variable, or vice-versa, respectively. Sites displaying such a dramatic evolutionary change may point to a loss or gain of function at the protein site, reflecting adaptation, or they may indicate epistatic interactions among sites. Here, we analyzed full genomes of HIV and SIV-1 and identified 271 rate-shifting sites along the HIV-1/SIV phylogeny. The majority of rate shifts occurred at long branches, often corresponding to cross-species transmission branches. We noted that in most proteins, the number of rate accelerations and decelerations was equal, and we suggest that this reflects epistatic interactions among sites. However, several accessory proteins were enriched for either accelerations or decelerations, and we suggest that this may be a signature of adaptation to new hosts. Interestingly, the non-pandemic HIV-1 group O clade exhibited a substantially higher number of rate-shift events than the pandemic group M clade. We propose that this may be a reflection of the height of the species barrier between gorillas and humans versus chimpanzees and humans. Our results provide a genome-wide view of the constraints operating on proteins of HIV-1 and SIV.

## 1. Introduction

The human immunodeficiency viruses HIV-1 and HIV-2 are the causative agents of AIDS in humans, infecting millions of people worldwide. Both viruses emerged from a clade of lentiviruses known as the simian immunodeficiency viruses (SIV), which naturally infect a variety of non-human primate species. HIV in humans arose from several independent transmission events of primate SIVs that resulted in HIV-1 groups M and N (from SIV infecting chimpanzees, SIVcpz), HIV-1 groups O and P (from SIV infecting gorillas, SIVgor), and HIV-2 groups A through H (from SIV naturally infecting sooty mangabeys, SIVsmm) [1,2,3]. The gorilla-infecting lentivirus, SIVgor, is itself a result of a transmission of SIVcpz from chimpanzees to gorillas [4,5]. Phylogenetic analyses date the most common recent ancestors of HIV groups M and O to the beginning of the 20th century, making it a relatively new human pathogen [6,7]. Similar analyses of SIVgor date the most recent common ancestor of this virus in gorilla somewhere in the 19th century [4]. SIVcpz itself was found to be a transmission from other primates, leading to two lineages of SIVcpz: SIVcpz*ptt* infecting the chimpanzee subspecies *Pan troglodytes troglodytes* of central Africa, and SIVcpz*pts* infecting the *Pan troglodytes schweinfurthii* chimpanzee subspecies of eastern Africa [5,8,9,10,11,12].

How viruses are able to cross the species barrier is a subject of much interest, since many pandemic human viruses arose from zoonosis events: the influenza strain of Spanish flu H1N1, Ebola Virus, SARS, and SARS-CoV-2 coronaviruses [13,14,15]. Due to differences between species, virus adaptation to new host species occurs at multiple stages in the virus infection cycle. To date, studies have found that the Gag, Env, and in particular, the Vpu protein of HIV likely underwent adaptation during the transition to a human host [16,17,18,19,20,21,22,23].

Protein adaptations are reflected in the history of genomes and may be manifested as changes in the rate of evolution at the adapting sites, known as rate shifts. For example, if a site was under weak constraint in the past (and exhibited a fast rate of evolution), following acquisition of a new function, the site will be under a higher degree of constraint (and will exhibit slower rate of evolution) [24,25,26,27]. However, recent work has shown that pervasive epistatic interactions among protein sites may also lead to observed changes at the rate of evolution at a site [28,29]. Accordingly, the presence of a certain amino acid at one site may lead to the tolerance of various amino acids at a second site and, hence, potentially a high rate of evolution at this second site. However, a substitution at the first site to another amino-acid may dictate loss of tolerance at the second site, also known as entrenchment, and hence, a lower rate of evolution [30,31,32,33]. Such epistatic patterns have indeed been found for the Env protein of HIV-1 [34,35].

Until today, the lack of SIVcpz and group O full genomic sequences limited the ability to infer rate shifts across cross-species transmission events. Here, we utilized the growing availability of diverse and full HIV-1 and SIV genomes to identify sites in SIVcpz/SIVgor/HIV-1 clades whose evolutionary rate significantly changed across clades. We next attempted to characterize the driving force behind the pattern of rate shifts and to infer whether adaptation or epistatic interactions characterize protein evolution across the HIV-1/SIV phylogeny.

## 2. Methods

In order to collect sequences for this study, the Los Alamos HIV sequence database (available online at http://www.hiv.lanl.gov [36], retrieved at 17 April 2016) was queried for HIV-1 sequences from the same strain that spanned all nine HIV-1 open reading frames (ORFs: *gag*, *pol*, *vif*, *vpr*, *tat*, *rev*, *vpu*, *env*, and *nef*) and for SIVcpz and SIVgor strains that spanned the corresponding ORFs (with the exception of *vpx*). This led to 2004 sequences of HIV-1 group M, 45 sequences of HIV-1 group O, 9 sequences of HIV-1 group N, 2 sequences of HIV-1 group P, 4 sequences of SIVgor, and 29 sequences of SIVcpz. The final study dataset was constructed with an equal number of group M and group O sequences (*N* = 45), by sampling sequences so that the *n* most distant strains (in terms of genetic distance) were sampled. Due to extremely high similarity, HIV-1 groups N and P sequences were reduced to a single representative strain from each. The IIIB_LAI strain was added manually as a reference sequence.

Initial MSA’s of the nine proteins were performed using PRANK and iteratively improved until convergence [37]. In order to reconstruct of the phylogenetic relationship between the sequences, we concatenated the alignments of Gag, Pol, Vif, Vpr, Tat, and Env and provided this as input for PhyML ([38], Appendix A). We next used the reconstructed phylogeny as a guide tree to realign each ORF with PRANK (Appendix A). JpHMM was used to validate that the strains used in the analysis are not inter-group recombinants [39].

In order to identify evolutionary rate shifts, we used RASER [25] to analyze each of the nine proteins separately, with the proteome-based phylogeny as input. RASER is a likelihood-based phylogenetic method for detecting a change in the substitution rates in each site of a given protein, along all possible branches in the phylogeny. First, a likelihood ratio test against a null model of no rate shifts is performed in order to assess if a model enabling rate shifts better fits the data. Next, the posterior probability of rate shift is calculated at each site and sites with a posterior probability higher than 0.6 are reported here. Finally, for each such site, the method lists the lineages where the rate shift occurred with the highest probability and further categorizes each sites as undergoing either a rate deceleration or a rate acceleration. All sites are reported in Appendix A.

Rate-shift events along the HIV/SIV phylogeny were constructed as the percentage of total rate-shift events at each branch, divided by protein size (Table 1). Another table containing the same rate shifts as percentage from total rate shifts at each branch, without taking into account the protein size, is present as Appendix A.

## 3. Results and Discussion

### 3.1. Finding Evolutionary Rate Changes

We have previously developed a maximum-likelihood-based method called RASER that identifies sites that undergo a significant change in their rate of evolution along a given branch of a phylogenetic tree (see for example Figure 1) [25]. RASER is based on two stages: first, it tests if a rate-shifting model better fits the data than a non-rate-shifting model based on a likelihood ratio test. Second, RASER reports the posterior probability that a protein site evolves under to a rate-shifting model and infers the branch in the phylogeny where the rate shift most likely occurred. Sites are said to be “rate decelerating” or “rate accelerating” at a given branch, if their evolutionary rate is, respectively, slower or faster in one clade of the tree as compared to the complementary clade spanned by the branch. For example, given the branch that separates HIV group M from SIVcpz, a site may be conserved (slower evolutionary rate) in the HIV group M clade and variable (faster evolutionary rate) in the rest of the tree. Notably, these terms are always relative: a rate deceleration at one clade can also be interpreted as rate acceleration in the complementary clade. Here, we sought to identify and characterize rate-shifting sites in the HIV/SIV phylogeny. We queried the Los Alamos HIV database [36] for all available HIV-1, SIVcpz and SIVgor full genome sequences. Rate-shift analyses were performed on the nine translated open reading frames of the virus, and likelihood ratio tests were found to be highly significant in favor of a rate-shift model across all HIV proteins (Appendix A).

### 3.2. Most Rate-Shift Events Are Identified in Long Branches

RASER identified a total of 271 rate-shifting sites along the HIV-1/SIV phylogeny. One hundred and thirty-seven of them were identified as rate deceleration events, and an additional 134 were identified as rate accelerations. Of the 271 rate-shifting sites, 230 were attributed to four branches: the branch separating group M from SIVcpz, the branch separating group O from SIVgor, the branch separating SIVgor and group O from SIVcpz, and the branch separating SIVcpz*pts* from other viruses (Figure 2B). Notably, these branches are the longest branches in the phylogeny and at least two of them represent cross-species transmission events.

In order to test for sequence sampling effects on the identified rate-shift patterns, we repeated the analysis with increased amount of group M sequences (*n* = 183), reduced amount of group O sequences (*n* = 13), and no SIVgor sequences (Appendix A). With this dataset, more rate decelerations were identified in the branch leading to group M, but chi-squared tests for differences in group M rate-shift distributions showed no significant difference (*p* = 0.11 and 0.87 for rate decelerations and accelerations, respectively), indicating that the patterns of rate shifts between the large group M sample and the smaller group M sample are similar.

### 3.3. Different Rate-Shift Patterns Identified in Different Branches

We calculated the relative contribution of rate-shift events in each protein when controlling for protein size, for each branch corresponding to a virus speciation event (Figure 2B). While rate-shift patterns differed between the different lineages, one common theme was that, at all prominent branches, a large fraction of rate-shift events occurred at non-structural proteins, i.e., the regulatory and accessory proteins of the virus (Table 1 and Appendix A). When comparing rate accelerations with the corresponding rate decelerations, the branch separating SIVcpz*pts* from the rest of the tree showed a very similar pattern of rate accelerations/decelerations: a very similar number of rate decelerations and rate accelerations were found in each protein (Figure 2A). A similar pattern was observed in the structural genes: a similar number of sites were inferred as rate accelerations and as rate decelerations in the Gag, Pol, and Env proteins (Table 1).

The pattern of equal accelerations and decelerations led us to conjecture that epistatic interactions coupled with genetic drift may be the most likely explanation for the majority of observed rate shifts. This is based on the fact that (a) a priori, we do not expect any adaptation in the branch separating SIVcpz*pts* from other viruses, and thus this branch serves as a reference for what to expect under neutral or almost neutral evolution and (b) strong epistatic effects have already been reported in Env [35], which indeed showed large and more or less equal numbers of accelerations and decelerations in most branches (apart from group O).

After establishing a pattern that seemed to be most likely neutral evolution, we went on to focus on outliers in our data: specific proteins where the number of rate accelerations and rate decelerations were substantially different. We noted higher proportions of rate decelerations as compared to rate accelerations in the Vpu protein, in the lineages leading to group M and group O, and in the O-Rev protein. Conversely, we noted more rate accelerations in Nef in the lineage leading to group M. Finally, we observed slightly higher proportions of rate decelerations in Vif and accelerations in Vpr in the lineage leading to SIVgor and group O.

### 3.4. Rate Shifts and Innate Immunity

We were intrigued by the fact that most of the imbalances found in rate accelerations/decelerations occurred in accessory proteins, which are known to play a role in the response of the virus to the innate immune response, and we survey some of our findings on rate shifts in the context of the interaction between innate immune proteins and HIV proteins.

One such well-characterized interaction is demonstrated by the restriction factor Tetherin, a fast evolving protein that differs between chimpanzees, gorillas, and humans [40,41]. In humans, a significant deletion in the cytoplasmic tail of Tetherin rendered it invulnerable to the SIV’s Nef-based counteraction [21], and in HIV-1 groups M and N, the Vpu protein adapted to counteract Tetherin through its transmembrane domain [22]. In gorillas, the Nef protein adapted to counteract its host’s Tetherin but also maintained its ability to counteract the chimpanzee Tetherin [42,43].

Our results agree with the described anti-Tetherin adaptation events: a high proportion of group M rate decelerations were found in the Vpu protein of group M, most of them in the identified Tetherin-binding domain (Table 1 and Figure 3). In addition, in HIV-1 group M the Nef protein was identified with three rate accelerations, which could be interpreted as relaxation of the SIV-Nef anti-Tetherin activity. Numerous group O Nef rate acceleration events were found as well, possibly indicating the loss of chimpanzee/gorilla anti-Tetherin function. A preponderance of acceleration and deceleration events in Nef were also found in the lineages leading to the ancestor of SIVgor/group O, and to SIVcpz*pts*. Thus, our analysis pinpoints sites that may be responsible for the crucial changes in function of Vpu and Nef throughout SIVcpz and HIV evolution (Appendix A).

We move on to examine APOBEC3G (A3G), a broad-range antiviral protein that is counteracted by the Vif proteins of Lentiviruses [45,46]. A3G variants of human, chimpanzee, and gorilla have been studied, and it has been found that the Vif recognition domain is conserved between human and chimpanzee A3G proteins, but slightly differs in gorillas [47]. Indeed, the SIVgor adaptation event to the different host A3G has been demonstrated [48,49]. In line with these results, our analysis shows no enrichment of rate decelerations in M-Vif, yet an excess of rate decelerations in the lineage of HIV-1 group O + SIVgor (Appendix A).

O-Rev protein stood out for an excess of rate decelerations in the branch leading to group O (Table 1). This is intriguing, since Rev is not known to be involved in any antiviral-related host activity. Some but not all of the decelerations identified are in genomic loci where *rev* rate decelerations overlap with *env* rate decelerations, and this might explain some of this pattern. It remains to be further investigated why the O-Rev protein experienced so many rate decelerations.

### 3.5. Discussion

Our results tentatively indicate that accessory proteins may have an important role in enabling adaptation to new species in the case of HIV [12]. This suggests that a major common barrier for a host species jump in HIV/SIV is the stage where the virus must overcome intracellular host defenses. Our results further support a lineage-specific model of human adaptation: groups M and O underwent prominent rate shifts in different proteins. This model is supported by the fact that activity of group M and O proteins are indeed different (e.g., Vpu) [42,43], and by the fact that each group originated from an SIV from a different primate.

We were surprised to note that group O had almost twice the amount of rate-shift events as compared to any other lineage analyzed (Table 1). Notably, group O is a non-pandemic strain that remained localized mainly to infections in west-central Africa [50,51]. One possible explanation is that group O entered the human population later than group M, and accordingly, more sites in group O appear as conserved, since they had less time to evolve. We consider this explanation as unlikely, since (a) our phylogenetic model takes into account branch lengths and (b) in such a case, we would also expect less rate accelerations in group O than in group M, whereas we find a very large number of rate accelerations as well. Another possible interpretation may be the relative height of the species barrier a strain had to overcome, as chimpanzees from which group M originated are genetically closer to human than gorillas from which group O originated [52,53]. A species barrier may potentially induce more entrenchment and may also lead to adaptation, as has been described previously [5,54].

### 3.6. Limitations

This study has several limitations. First, the availability of fully sequenced SIVcpz, SIVgor, and non-HIV-M genomes is still low, reducing the statistical power of the analysis. Indeed, we noted that increasing the sample size of group M sequences that are much more available led to the detection of more rate-shifting sites. We expect that the availability of additional SIVgor sequences will increase the number of rate-shift events that are unique in that clade, possibly revealing more rate changing sites for this lineage. Second, the method that we utilized to identify rate shifts is calibrated to identify major changes in the evolutionary rates along a lineage. Accordingly, it cannot detect sites where a “content shift” occurred, i.e., the rates of substitution remained low in two complementary clades, but the amino acid itself has changed. This is partially demonstrated in Gag_30_, a previously found marker of HIV adaptation where the rate of evolution changed mildly, while the amino acid in this site changed between chimpanzee and human viruses (Appendix A) [18]. Accordingly, we have detected this site but were inconclusive regarding the lineage where the rate shift occurred, and thus, it was not reported in this study. In addition, our method could not detect adaptation by amino acid insertion or deletion events, and we did not analyze for rate shifts at the RNA level and thus, for example, did not detect rate shifts at RNA structures of the virus.

## 4. Conclusions

We used a phylogenetic modeling approach to perform a large scale analysis of the constraints operating in all proteins of the HIV genome, across different lineages of HIV-1 and SIV. We inferred individual amino acids where rate shifts occurred and characterized the distribution of rate shifts across different proteins and across the tree. We found that, in many cases, the number of rate accelerations and rate decelerations is similar and suggest that epistatic interactions are pervasive in the evolution of HIV-1 and SIV proteins, and that these likely led to many of the inferred rate shifts observed herein. Nevertheless, our results do suggest that at least some of the rate shifts that we detected may be due to selection and may be related to the cross-species transmission of the virus from apes into humans. Our results also implicate several of the accessory proteins of HIV as mediators that allow cross-species transmission events to occur. Our compiled list of the positions suggested as rate shifting can now serve as a basis for future functional studies that will allow a better and deeper understanding of the constraints operating on HIV-1 and SIV proteins.

## Figures and Tables

**Figure 1 viruses-12-01312-f001:**
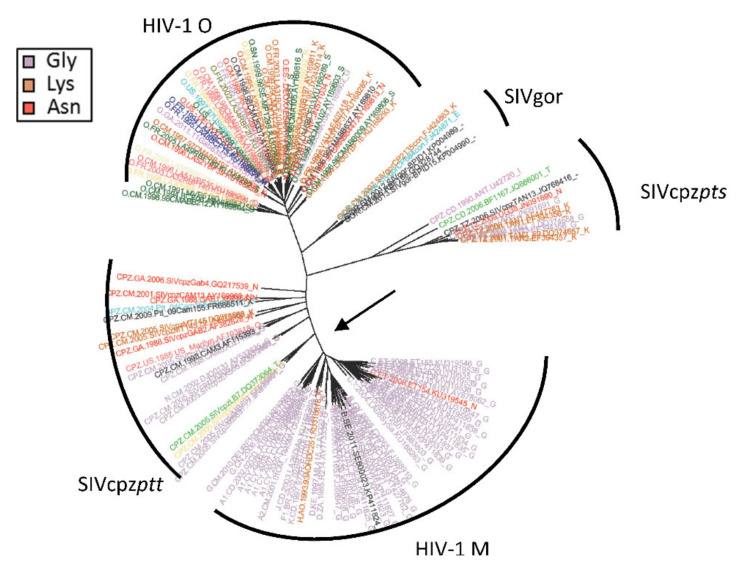
Projection of Env_458_ on the HIV-1/SIV phylogeny. Each leaf (corresponding to an HIV/SIV strain) is color-coded based on the amino acid present at Env_458_. The legend shows the three most prevalent amino acids found at this site. The substitution rate of Env_458_, which interacts with the CD4 receptor, was found to be slower in HIV-1 group M than in the rest of the phylogeny, with the branch separating both clades marked by an arrow.

**Figure 2 viruses-12-01312-f002:**
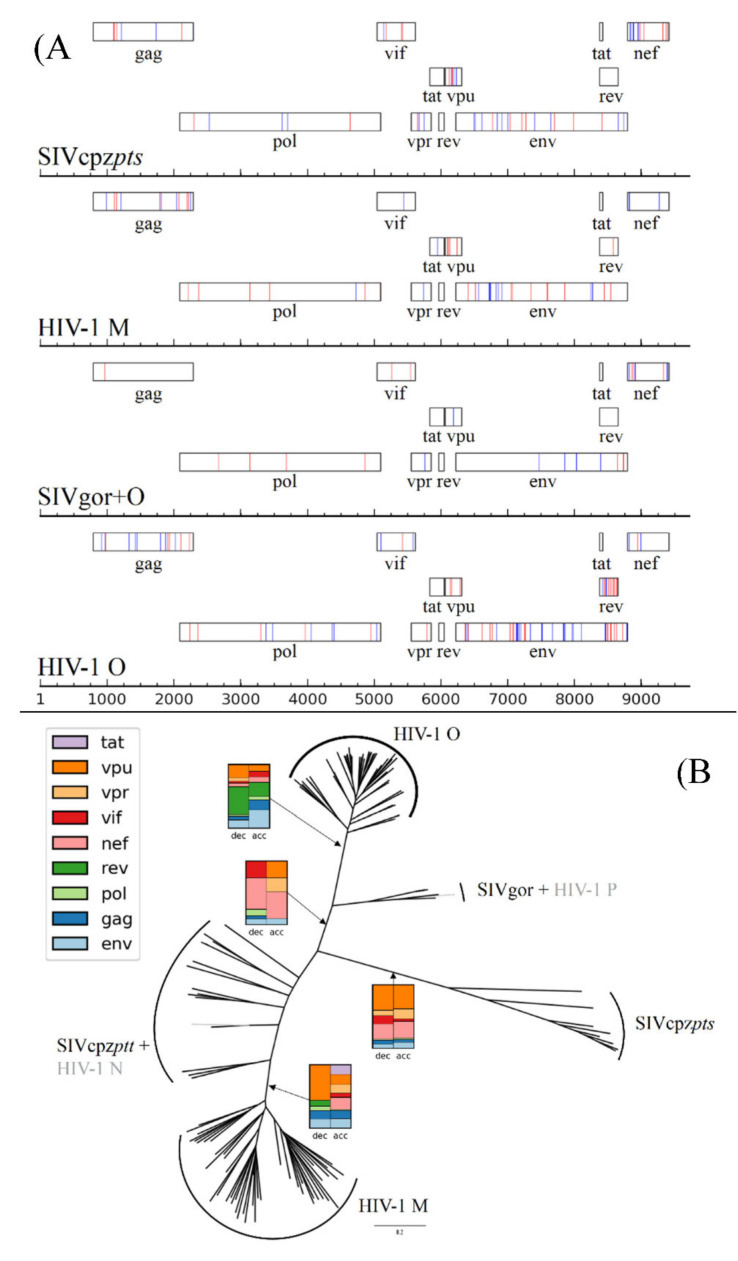
(**A**) Proposed rate deceleration (red) and acceleration (blue) events in prominent branches shown along the genome using the coordinates of the HXB2 reference strain. (**B**) Proposed rate-shift patterns in prominent branches, shown on the phylogeny. Rate shifts (deceleration left and acceleration right) are shown as the percentage of total rate-shift events at each branch, while controlling for protein size.

**Figure 3 viruses-12-01312-f003:**
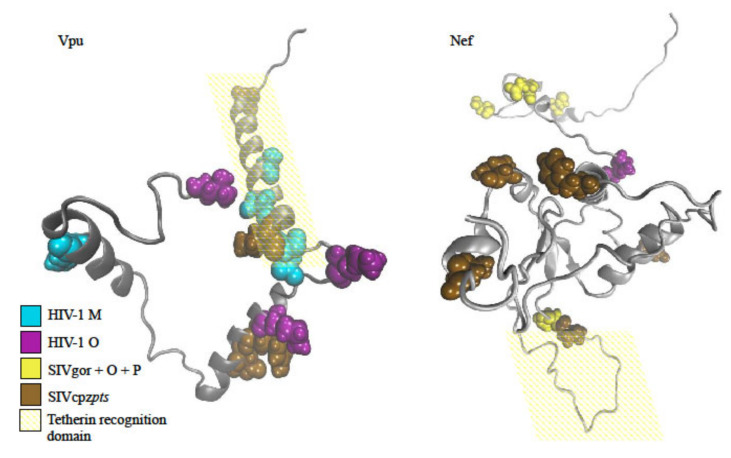
Projection of the identified rate decelerating sites for the prominent branches onto the structure of Vpu (**left**, PDB ID: 2N28) and Nef (**right**, obtained from [44]).

**Table 1 viruses-12-01312-t001:** The proportion of rate shifts (rate decelerations and rate accelerations in upper panel and lower panel, respectively) detected in each of the major branches of the phylogeny, relative to the protein size. The right-most column shows the absolute number of rate shifts detected in the branch. Cell background color intensity increases with the proportion of rate shifts.

	Structural Proteins	Regulatory Proteins	Accessory Proteins	
DEC Branch/Protein	Env	Gag	Pol	Rev	Tat	Nef	Vif	Vpr	Vpu	Total Number of Rate Shifts
Group M	1%	1%	1%	1%	0%	0%	0%	0%	5%	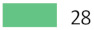
Group O	2%	1%	0%	8%	0%	1%	1%	1%	4%	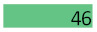
SIVgor + P + O	0%	0%	0%	0%	0%	2%	1%	0%	0%	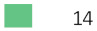
SIVcpz*pts*	1%	1%	0%	0%	0%	3%	2%	1%	5%	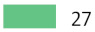
ACC Branch/Protein	Env	Gag	Pol	Rev	Tat	Nef	Vif	Vpr	Vpu	Total Number of Rate Shifts
Group M	1%	1%	0%	0%	1%	1%	1%	1%	1%	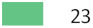
Group O	3%	2%	1%	3%	0%	1%	2%	0%	1%	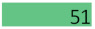
SIVgor + P + O	0%	0%	0%	0%	0%	2%	0%	1%	1%	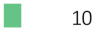
SIVcpz*pts*	1%	1%	0%	0%	0%	3%	1%	2%	5%	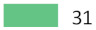

DEC = rate decelerations; ACC = rate accelerations.

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
