# Peer review of "Site-Specific Evolutionary Rate Shifts in HIV-1 and SIV"

_viruses, 2020, doi:10.3390/v12111312_

Round 1

Reviewer 1 Report

This is an interesting study which has several limitations, however these issues are noticed by the authors in Limitations section. I found specially interesting the section when the authors discuss about rate shifts and its possible effect in innate immunity. 

Minor changes

The last sentences of Methods section (comparison large group M vs. small group M) report dates (p values) that maybe could be better suitable in Results section. 

Author Response

Reviewer #1:

This is an interesting study which has several limitations, however these issues are noticed by the authors in Limitations section. I found specially interesting the section when the authors discuss about rate shifts and its possible effect in innate immunity. 

Minor changes

The last sentences of Methods section (comparison large group M vs. small group M) report dates (p values) that maybe could be better suitable in Results section. 

Response: Thanks, this paragraph was moved into the results section (lines 135-141).

Reviewer 2 Report

Gelbart and Stern have used an evolutionary modeling of rate shifts and report that several accessory proteins are enriched for rate shifts, suggesting that SIVs have evolved to overcome species-specific anti-viral proteins to adapt to new hosts. Although this report offers some insights into virus-host interaction especially upon a zoonotic transmission, there are several points that need to be addressed in order to reveal the relevance of these findings. Some specific questions and concerns are the following.

One mysterious thing is that there are not many changes in Vif. APOBEC proteins are one of the most dominant host restriction factors against lentiviruses, and there are always evolutional conflicts between these proteins and viral Vif proteins. In fact, SIVcpz had to evolve its Vif to antagonize APOBEC proteins of gorillas. The authors should discuss this point.

The authors should discuss the changes in Rev. They find a significant number of changes in the protein in the group O branch. There is no evidence that Rev antagonizes known host restriction factors, and one may wonder why Rev needs to evolve that much.

As the authors also point out, the limited number of strains that can be tested for the SIVgor-HIV-1 P branch may need further discussion. From the current analysis, they form a single group, implying there was only a little evolutional changes when SIVgor had jumped into human population. This scenario is hard to believe given the genetic differences between two species. Would we see more differences in rate shifts between SIVgor and HIV-1 P as we discover more strains? The authors may be able to make some additional comments.

Author Response

Reviewer #2:

Gelbart and Stern have used an evolutionary modeling of rate shifts and report that several accessory proteins are enriched for rate shifts, suggesting that SIVs have evolved to overcome species-specific anti-viral proteins to adapt to new hosts. Although this report offers some insights into virus-host interaction especially upon a zoonotic transmission, there are several points that need to be addressed in order to reveal the relevance of these findings. Some specific questions and concerns are the following.

One mysterious thing is that there are not many changes in Vif. APOBEC proteins are one of the most dominant host restriction factors against lentiviruses, and there are always evolutional conflicts between these proteins and viral Vif proteins. In fact, SIVcpz had to evolve its Vif to antagonize APOBEC proteins of gorillas. The authors should discuss this point.

Response: Thanks, text was added at lines 181-182:

"…and we survey some of our findings on rate shifts in the context of the interaction between innate immune proteins and HIV proteins."

And the following paragraph was added to discuss this in lines 194-200:

"We move on to examine APOBEC3G (A3G), a broad-range antiviral protein that is counteracted by the Vif proteins of Lentiviruses [45,46]. A3G variants of human, chimpanzee and gorilla have been studied, and it has been found that the Vif recognition domain is conserved between human and chimpanzee A3G proteins, but slightly differs in gorillas [47]. Indeed, the SIVgor adaptation event to the different host A3G has been demonstrated [48,49]. In line with these results, our analysis shows no enrichment of rate-decelerations in M-Vif, yet an excess of rate decelerations in the lineage of HIV-1 group O + SIVgor (Tables S2, S3)."

The authors should discuss the changes in Rev. They find a significant number of changes in the protein in the group O branch. There is no evidence that Rev antagonizes known host restriction factors, and one may wonder why Rev needs to evolve that much.

Response: This is indeed a mystery, which may partially be explained by rate shifts in the overlapping Env protein. The following text was added (lines 201-205):

" O-Rev protein stood out for an excess of rate-decelerations in the branch leading to group O (Table 1). This is intriguing since Rev is not known to be involved in any antiviral-related host activity. Some but not all of the decelerations identified are in genomic loci where rev rate decelerations overlap with env rate decelerations, and this might explain some of this pattern. It remains to be further investigated why the O-Rev protein experienced so many rate decelerations."

As the authors also point out, the limited number of strains that can be tested for the SIVgor-HIV-1 P branch may need further discussion. From the current analysis, they form a single group, implying there was only a little evolutional changes when SIVgor had jumped into human population. This scenario is hard to believe given the genetic differences between two species. Would we see more differences in rate shifts between SIVgor and HIV-1 P as we discover more strains? The authors may be able to make some additional comments.

Response: We don't expect seeing more rate shifts in the HIV-1 P group since it did not spread within the human population – there are only 3-4 known sequences and they are all extremely similar. We cannot test for rate-shifts under such a scenario. We now write in the text: “This study has several limitations. First, the availability of fully sequenced SIVcpz, SIVgor, and non HIV-M genomes is still low, reducing the statistical power of the analysis.”